# A Quantum Ring Laser Gyroscope Based on Coherence de Broglie Waves

**DOI:** 10.3390/s22228687

**Published:** 2022-11-10

**Authors:** Byoung S. Ham

**Affiliations:** School of Electrical Engineering and Computer Science, Gwangju Institute of Science and Technology, 123 Chumdangwagi-ro, Buk-gu, Gwangju 61005, Korea; bham@gist.ac.kr

**Keywords:** Sagnac interferometer, ring laser gyroscope, quantum coherence, coherence de Broglie waves, sensing

## Abstract

In sensors, the highest precision in measurements is given by vacuum fluctuations of quantum mechanics, resulting in a shot noise limit. In a Mach–Zenhder interferometer (MZI), the intensity measurement is correlated with the phase, and thus, the precision measurement (Δn) is coupled with the phase resolution (Δφ) by the Heisenberg uncertainty principle. Quantum metrology offers a different solution to this precision measurement using nonclassical light such as squeezed light or higher-order entangled-photon pairs, resulting in a smaller Δφ and sub-shot noise limit. Here, we propose another method for the high precision measurement overcoming the diffraction limit in classical physics, where the smaller Δφ is achieved by phase quantization in a coupled interferometric system of coherence de Broglie waves. For a potential application of the proposed method, a quantum ring laser gyroscope is presented as a quantum version of the conventional ring laser gyroscope used for inertial navigation and geodesy.

## 1. Introduction

Precision measurements are at the heart of sensing and metrology [1,2,3,4,5,6,7,8]. In statistics, a standard deviation is proportional to the square root of the number of measurements. The minimum sensitivity of the shot noise in classical physics is caused by the uncertainty principle of quantum physics. This is called a standard quantum limit, which determines the sensitivity limit in classical physics. On the other hand, the diffraction limit or Rayleigh criterion classically determines the maximum resolution of sensors. Thus, multi-wave interference in an optical cavity is a typical method to enhance the resolution limit satisfied by coherence optics. In contrast to classical physics, quantum mechanics offers a quantum advantage in sensing, imaging, and metrology, where higher-order entangled photon pairs play a major role in overcoming the standard quantum limit by a factor of the square root of N, where N is the total number of photons in the entangled pair [3,4,5,6,7,8]. The order of an entangled photon pair is represented by the N number of photonic de Broglie waves (PBWs) [9,10,11]. Due to the indeterminacy and difficulties of PBW generations, however, the implementation of quantum sensing for applications such as lithography [11], frequency standards [12], imaging [13], and spectroscopy [14] has been severely limited.

Quantum mechanics is rooted in the wave-particle duality [15]. Unlike PBWs based on the particle nature of a photon, the wave nature of coherence de Broglie waves (CBWs) [16,17] has been recently investigated for quantum sensing applications [18]. The physics of CBWs is in the phase-basis superposition between coupled MZIs [16,17]. Owing to the on-demand control of the geometric scalability of MZIs and the inherent benefit of a single-shot measurement, CBWs provide new opportunities for quantum sensing to overcome the limitations in both quantum and classical counterparts. Such a quantum feature of CBWs can be applied for various quantum engineering fields of sensing and metrologies. Recently, the first application of CBWs to sensors has been proposed for a CBW Sagnac interferometer, whose resolution limit overcomes conventional counterparts based on cavity interferometers [18]. So far, the Sagnac gyroscope has been implemented for optical [19] and matter-wave [20] interferometry as well as atomic spectroscopy [21], gravitational wave detection [22], inertial navigation [23] and geodesy [24,25]. In particular, the ring laser gyroscope (RLG) offers a highest sensing (resolution) capability up to one part of $10^{8}$ in the earth’s rotation rate Ω [26,27]. Thus, in the RLG, enhancing phase resolution is at the heart of applications. Here, a quantum version of RLG is presented using CBWs, whose sensing capability in phase resolution overcomes the classical limit of RLG. Compared with the CBW Sagnac interferometer [18], the proposed quantum RLG is an active version, whose phase resolution is greatly enhanced compared with the classical counterpart of the RLG. Moreover, it can be applied directly to the RLG with a minimal modification of optical geometry.

## 2. Materials and Methods

Figure 1a shows a schematic of the proposed CBW-based quantum RLG. Figure 1b is the unfolded scheme of Figure 1a and shows two cavity modes. As in conventional RLGs, these two modes are independent. The two modes are, however, coherent within the cavity due to the shared path length and control parameters of the cavity. Figure 1c is an equivalent scheme of Figure 1a for one mode of either left or right directed light, where the modified region with green MZIs (non-shaded) plays the function of phase-basis quantization via superposition between consecutive MZIs (yellow region) across the BS (dotted region) in Figure 1a (see the green-dotted box) [16]. The green dotted region of Figure 1a with a nonpolarizing 50/50 beam splitter, a path-length controller (piezoelectric transducer 1, PZT 1), and a pair of cavity mirrors represents a modified scheme of RLG for the quantum RLG. Here, PZT 1 represents a control parameter of the phase φ to control the cavity length. PZT 2 is another control parameter of the quantum RLG, where the phase ζ is for the ring cavity condition with the asymmetric (counter propagating light-caused ±ψ; Sagnac effects) MZI configuration [16]. The ψ-asymmetric MZI configuration is automatically accomplished by the Sagnac effect for the counter-propagating fields [28,29].

Compared with the original CBW scheme [16], this ±ψ configuration is an essential part of the proposed scheme. Compared with the CBW Sagnac interferometer [18], Figure 1 is an active version of it with an optical gain [30,31]. Here, the optical gain has no direct relation to the enhanced phase resolution due to their independence. In each round of circulation by the dotted region in Figure 1a, the CBW mode increases linearly [16,17]. In an ideal optical cavity, this mode goes to infinity. Unlike the Fabry–Perot (FP) interferometer in conventional RLGs, however, the wavelength $\lambda_{CBW}$ of CBW is linearly doubled. Thus, the optical gain of each ordered CBW is just for a one time pass in each mode. Instead, all CBWs are linearly superposed resulting in interference in the cavity. Detailed discussions of the nonlinear effects in the gain medium are beyond the scope of the present paper. Due to the gain-induced high signal-to-noise ratio (SNR), the optical gain should affect the sensitivity in a classical regime. Analysis of quantum sensitivity is also beyond the scope of the present research. The dithering [30] in RLG due to the imperfect mirror scattered coupling with its original light can also be studied elsewhere, but may not be effective due to different CBW modes.

For the numerical simulations in Figure 2 and Figure 3, we have made a Matlab program for two dimensional calculations of both output fields’ amplitudes in Equation (2). The CBW order m is practically set with respect to the cavity Finesse, where the reflection coefficient ‘r’ of the mirror plays an important role. For this, both parameters of m and ψ are varied, where the m-based ordered fields $E_{A}^{\left(m\right)}$ and $E_{B}^{\left(m\right)}$ are linearly added for each ψ value for the final $E_{A}$ and $E_{B}$. For this, the one-time pass gain effect is added. The increment step of ψ is set at 0.0001π. Finally, the output intensity $I_{A}$ ($I_{B})$ is obtained via conjugate products of $E_{A}$ and $E_{A}^{*}$ ($E_{B}$ and $E_{B}^{*}$). For the mechanical noise effects on CBWs, a Mabtlab commend rand (1) is used for random number generation in Figure 3b.

## 3. Results

According to the original CBWs [16], the basic building block is composed of the green-yellow MZIs as denoted by ‘p’ number [17] (see Figure 1c). In the yellow MZIs, however, PZT 2 (ζ) (see Figure 1a,b) caused by environmental noises such as vibrations, temperatures, and air turbulences does not affect the Sagnac effect due to exact phase cancelation by the counter-propagating fields. Such a self-phase maintenance has been experimentally demonstrated in a quantum version of the Sagnac interferometer [32]. In the modified region (see the green-dotted box in Figure 1a), there is no net Sagnac effect, either, due to the geometrical symmetry. Compared with the Sagnac scheme of [32], the double unitary transformation of CBWs gives a much subdued phase stabilization, whose phase noise is from the phase difference between forward and backward MZIs [33]. Thus, the Sagnac effect in the proposed scheme has a great benefit of environmental noise reduction compared with RLG [23,24,25]. As a result, any rotation rate Ω induces a time delay between the counter-propagating fields inside the ring, resulting in the Sagnac effect as in the original passive version of CBW-RLG [18]. This is the passive form of CBW-RLG, where the Sagnac effect appears as a phase shift. In the present active form of CBW-RLG with an embedded gain medium L in Figure 1a, the rotation-caused phase shift in the passive CBW-RLG now appears as a frequency shift under a lasing mode [34]. Between them, i.e., the phase shift $\left(\varphi \right)$ in the passive RLG and the frequency shift $\left(∆\omega \right)$ in the active RLG, an exactly equivalent relation is satisfied [23]: $∆\omega =\varphi \frac{c}{L}$, where c is the speed of light and L is the perimeter of a ring cavity. Meanwhile, the Ω-induced Sagnac effect is neglected in the following analysis of the proposed stand-still quantum RLG for simplicity to prove the quantum gain in phase resolution.

Using coherence optics of a BS [35], the output fields A and B in Figure 1a for the round trip of the optical cavity are obtained via matrix representations for Figure 1c as follows:(1)$$\left[\begin{array}{c} E_{A}\\ E_{B} \end{array}\right]={\left[MZI\right]}_{+}\left[\varphi \right]{\left[MZI\right]}_{-}e^{\eta }e^{i\xi }\left[\begin{array}{c} E_{0}\\ 0 \end{array}\right]={\left(-1\right)}^{1}e^{i\xi }E_{0}e^{\eta }\left[\begin{array}{c} \cos\psi \\ \sin\psi \end{array}\right]$$
where ${\left[MZI\right]}_{+}=\left[BS\right]\left[\psi^{+}\right]\left[BS\right]=\frac{1}{2}\left[\begin{array}{cc} 1-e^{i\psi } & i\left(1+e^{i\psi }\right)\\ i\left(1+e^{i\psi }\right) & -\left(1-e^{i\psi }\right) \end{array}\right]$ and ${\left[MZI\right]}_{-}=\left[BS\right]\left[\psi^{-}\right]\left[BS\right]=\frac{1}{2}e^{i\xi }\left[\begin{array}{cc} 1-e^{-i\psi } & i\left(1+e^{-i\psi }\right)\\ i\left(1+e^{-i\psi }\right) & -\left(1-e^{-i\psi }\right) \end{array}\right]$, respectively. Here, $\left[BS\right]=\frac{1}{\sqrt{2}}\left[\begin{array}{cc} 1 & i\\ i & 1 \end{array}\right],\;\left[\psi^{+}\right]=\left[\begin{array}{cc} 1 & 0\\ 0 & e^{i\psi } \end{array}\right],$ and $\left[\psi^{-}\right]=\left[\begin{array}{cc} 1 & 0\\ 0 & e^{-i\psi } \end{array}\right]$.The global phase ξ caused by the added region for quantum superposition (see the green-dotted box) is slowly varying. For simplicity, a laser gain in the ring cavity is denoted by $e^{\eta }$, where η is the gain coefficient for a round trip of each mode. This gain effect has no quantum influence in the phase resolution but may give a classical advantage, where $E_{0}$ is determined by the embedded gain medium L. The phase ψ is due to the rotation (Ω)-induced Sagnac effect, and $\left[\varphi \right]=\left[\begin{array}{cc} 1 & 0\\ 0 & 1 \end{array}\right]$ is set with φ=2nπ (*n* = 1, 2, 3, …) by controlling PZT 1. The PZT 2 is for the ring cavity, where the phase ζ is invariant to the Sagnac effect. Unlike the original CBWs [16], the ψ in the basic building block of Figure 1c is induced by the Sagnac effect. The global phase $e^{i\xi }$ in Equation (1) has no effect on measurements. For CBWs, however, φ=0 must be satisfied, otherwise $E_{11}=-e^{i\xi }E_{0}$ and $E_{12}=0$. Thus, the general solution of the *m*^th^ ordered CBW is as follows (*m* = 2p):(2)$${\left[\begin{array}{c} E_{A}\\ E_{B} \end{array}\right]}^{m}=e^{im\xi }e^{m\eta }E_{0}{\left[CBW\right]}^{m}\left[\begin{array}{c} E_{0}\\ 0 \end{array}\right]={\left(-1\right)}^{m}e^{im\xi }e^{m\eta }E_{0}\left[\begin{array}{c} \cos\left(m\psi \right)\\ \sin\left(m\psi \right) \end{array}\right]$$
where ${\left[CBW\right]}^{m}={\left({\left[MZI\right]}_{+}{\left[MZI\right]}_{-}\right)}^{m}={\left(-1\right)}^{m}{\left[\begin{array}{cc} cos\psi & -sin\psi \\ sin\psi & cos\psi \end{array}\right]}^{m}$, resulting in $E_{A}^{\left(m\right)}={\left(-1\right)}^{m}e^{im\xi }e^{m\eta }\cos\left(m\psi \right)E_{0}$ and $E_{B}^{\left(m\right)}={\left(-1\right)}^{m}e^{im\xi }e^{m\eta }\sin\left(m\psi \right)E_{0}$. This is the phase-basis quantization of CBWs with $\psi_{m}\in \left\{0,\frac{\pm \pi }{m}\right\}$ [17]. Here, the CBW order m is given by the round-trip number in Figure 1a. As mentioned in the Methods section, the gain is one time for each CBW mode, and thus the nonlinear optics of lasing are excluded.

Regarding the output intensities $I_{A}$ and $I_{B}$ in Figure 1a detected by D1 and D2, respectively, where the corresponding amplitudes are $E_{A}=E_{0}\sum_{m}{\left(-1\right)}^{m}e^{im\xi }e^{m\eta }\cos\left(m\psi \right)$ and $E_{B}=E_{0}\sum_{m}{\left(-1\right)}^{m}e^{im\xi }e^{m\eta }\sin\left(m\varphi \right)$, the following analytical solutions are obtained.

(i)$m\psi =\pm \frac{\left(2n-1\right)\pi }{2}$, where *n* = 1, 2, 3, …

For all m, $E_{A}^{\left(m\right)}=0$. $E_{B}^{\left(m\right)}=0$ is also satisfied due to the ${\left(-1\right)}^{m}$ effect. Thus, $I_{A}=I_{B}=0$.

(ii)mψ=±2nπ, where *n* = 1, 2, 3, …

For all m, $E_{A}^{\left(m\right)}=0$ at ψ=±2nπ due to the ${\left(-1\right)}^{m}$ effect, resulting in destructive interference. $E_{B}^{\left(m\right)}=0$ is automatically satisfied. Thus, $I_{A}=I_{B}=0$.

(iii)$m\psi =\pm \left(2n-1\right)\pi$, where *n* = 1, 2, 3, …

For all m, all $E_{A}^{\left(m\right)}$ is zero. Thus, $I_{A}=0$. However, $E_{B}^{\left(m\right)}$ constructively interferes at $\psi =\pm \left(2n-1\right)\pi$ due to the ${\left(-1\right)}^{m}$ effect, resulting in $I_{B}=e^{2m\eta }I_{0}$, where the η is the gain in each pass. These output features of Figure 1 are definitely different from conventional cavity optics of RLG (see Figure 2). Now, we need to clarify whether Equation (2) is rooted in coherence optics of multi-wave interference or quantum optics with the phase-basis quantization to show the novelty of the proposed quantum RLG.

For the detailed analysis, numerical calculations are as shown in Figure 2 for Equation (2). Figure 2a shows the normalized output intensity $I_{B}$ detected by D2 in Figure 1a. For the calculations, the gain coefficient $e^{m\eta }$ is not considered for simplicity for the minimum effect of the quantum advantage in phase resolution. As analyzed above in (i)–(iii), the constructive interference appears at $\psi =\pm \left(2n-1\right)\pi$ for $E_{B}$. This π-shifted fringe with respect to the conventional cavity optics is due to the inserted BS, as shown in Figure 1b, resulting in a π phase shift between two identical cavities. As analyzed $I_{A}=0$ is achieved at $\psi =\pm \left(2n-1\right)\pi$ (see the red curve of Figure 2b).

Figure 2b is an expanded (unfolded) version of Figure 2a for both output intensities, where $I_{A}\;\left(I_{B}\right)$ is denoted in red (blue). For comparison purposes, the green dotted curve which is π phase shifted shows the classical resolution limit of a Fabry–Perot (FP) interferometer. The enhanced phase resolution of $I_{B}$ with respect to the reference is at least three times. Thus, the nonclassical breakthrough in phase resolution is demonstrated, where the reference of FP is the classical limit in resolution of conventional RLG. Thus, Figure 1a shows the quantum advantage of the proposed quantum RLG, where the breakthrough in Figure 2b is due to quantum superposition of the phase quantized CBWs [17]. This quantum advantage cannot be obtained from classical physics.

Figure 2c,d illustrate destructive and constructive interferences in $E_{B}$. For this, some neighboring samples are shown for the $m^{\mathrm{th}}$ and ${\left(m+1\right)}^{\mathrm{th}}$ ordered amplitudes in $E_{B}^{\left(m\right)}$. From the symmetric distribution, the destructive interference at ψ=±2nπ in Figure 2c is due to the ${\left(-1\right)}^{m}$ effect in Equation (2). In contrast, there is a constructive interference at $\psi =\pm \left(2n-1\right)\pi$ in Figure 2d, as analyzed above: neighboring curves are overlapped.

Figure 2e,f show all ordered amplitudes up to *m* = 5000 as a function of ψ. As the order m increases, the amplitudes of both $E_{A}^{\left(m\right)}$ and $E_{B}^{\left(m\right)}$ decrease. Considering the cavity gain ($e^{m\eta }$), however, the m number can be increased for higher-order amplitudes (discussed in Figure 3), where this increasing effect may be equivalent to increasing reflection coefficient ‘r’, as usual. The sum of amplitudes for all modes of $E_{B}^{\left(m\right)}$ constructively interfere only at $\psi =\pm \left(2n-1\right)\pi$, as shown in Figure 2b. Due to the independency between two modes of the oppositely directed lights from the embedded gain medium L in Figure 1, there is no phase coupling by the other mode. Due to the shared cavity and coherently ordered CBWs at $\lambda_{CBW}$, no random phase fluctuations occur in $E_{A}^{\left(m\right)}$ and $E_{B}^{\left(m\right)}$.

Figure 3a shows contrast-based phase resolution change for Figure 2b. For this, the m proportional intensity decrease is assumed, resulting in less CBWs. From the reference with m = 5000 (blue curve), the collection efficiency decrease is represented for decreased CBW numbers to m = 2000 (red), m = 1000 (green), and m = 500 (black). This may also be related with decreased signal level close to the shot noise limit for the same cavity. Thus, the passive quantum ring gyroscope in [18] should result in less resolution only due to less collection efficiency in all possible CBWs in Figure 1.

Figure 3b is for the purpose of comparison between no-noise (red) and noise-allowed (green) ring cavities. The given phase noise is assumed to be random within ±0.2π, resulting in a blurry resolution (see the green band). This noise effect is of course classical. However, it can be reduced in the present CBW-based quantum ring gyroscope due to the double unitary transformation of CBWs. Theoretical analysis of noise cancellation has been recently conducted for an optical link of unconditionally secured classical key distribution in a CBW scheme [33].

Figure 3c is for the gain effect. For analytical consistency, the gain coefficient η compensates the cavity loss, resulting in increased m number in the same cavity. This effect is the same as increasing ‘r’ for the same m=5000 in Figure 3b. Compared with the red curve in Figure 3b, which is the average of the noise effect, an enhancement in resolution is demonstrated in the blue curve. This enhancement is of course classical.

## 4. Discussion

A general advantage of the active RLG compared with passive one is the better sensitivity in phase resolution owing to the lasing mode difference excited by counter-propagating light fields [23]. Due to different frequencies Δω caused by the Sagnac effects, it is well known that a heterodyne detection technique gives much better detection resolution compared with the phase-shift measurement in the passive form [23,30,31]. The disadvantage of the active RLG compared with a passive one is, however, is the size independence due to the perimeter-independent scale factor [23]. Thus, the ratio of RLG’s angular velocity Ω to the wavelength λ becomes inaccurate for small Ω due to the lock-in effect. In the present active design of CBW-RLG, the wavelength λ is replaced by $\lambda_{CBW}\;\left(=\frac{\lambda }{m}\right)$ effectively, where m is the unit of CBW as shown in Equation (2) and Figure 1c [16,17]. Thus, the RLG’s main disadvantage may be alleviated with this m factor in addition to the enhanced phase-resolution demonstrated in Figure 2. Due to the equivalence relation between the phase shift in a passive RLG and the frequency difference in the active RLG, the demonstration of breakthrough in phase resolution represents the same breakthrough in frequency resolution for heterodyne detection [23].

Compared with conventional diffraction-limited RLGs, the proposed quantum RLG is based on quantum superposition of CBWs in a modified ring cavity. Due to the embedded gain medium, two independent cavity modes exist, where in the present paper, only one mode is investigated. Regarding each mode of optical fields oppositely directed, both counter-propagating CBWs are generated in the modified region of Figure 1, resembling both modes of the conventional RLG. As shown in Figure 2b, thus, only one detector is sensitive to the CBW fringes. The opposite mode of light from the gain medium L in the cavity gives a swapped result according to MZI physics, where the input field direction is reversed to the side in Figure 1c. Thus, the sensing detector is also swapped from D2 to D1, in which both detectors show the same feature due to the input swapping. However, the intensity product between two detectors should be doubly enhanced in sensing compared with one detector capability as in the two-photon intensity correlation. To prove these analyses, experimental studies may be followed.

The Sagnac effect caused by counter-propagating fields in an optical cavity is denoted by the phase shift ψ for the present analysis. This phase shift can also be represented by the frequency difference between them, resulting in the normal beating signal in conventional RLG detection methods. In Figure 2 and Figure 3, however, the phase shift ψ has been used to demonstrate the enhanced phase resolution, i.e., the quantum advantage of the present method. The analytical tool of linear optics in Equations (1) and (2), and the following numerical calculations result in general solutions under the coherence optics of multimode CBWs. Unlike conventional cavity optics, CBW’s amplitude gain is for one time at each mode of CBWs, satisfying liner superposition. The sensitivity issue is beyond the scope of present research because it is a completely different matter from the phase resolution in quantum sensing [4].

## 5. Conclusions

In conclusion, an active quantum RLG was presented for enhanced phase-resolution capabilities overcoming the classical limits of RLG. The quantized CBW modes due to linear superposition of coupled MZIs in an optical cavity were analyzed for the quantum advantage in phase resolution, where the enhanced resolution was numerically confirmed. Compared with cavity optics-based conventional RLG, the proposed quantum RLG showed a quantum advantage in the phase resolution overcoming the diffraction limit three times greater than that of the RLG limit. However, this quantum gain can be enhanced more due to the cavity gain-caused amplitude increase in the higher order CBWs. Eventually, the general advantage of using a beating signal between two counter-propagating different lasing modes in the active RLG can be enormously enhanced due to the m factor in the effective wavelength $\lambda_{CBW}$ for a beating signal count. Although the CBW-RLG physics belongs to the many-wave interference as in coherence optics, the discrete frequency-ordered CBWs should behave differently from conventional cavity optics limited to the same frequency. Further research on sensitivity and nonlinear optics of the CBW-RLG will follow.

## Figures and Tables

**Figure 1 sensors-22-08687-f001:**
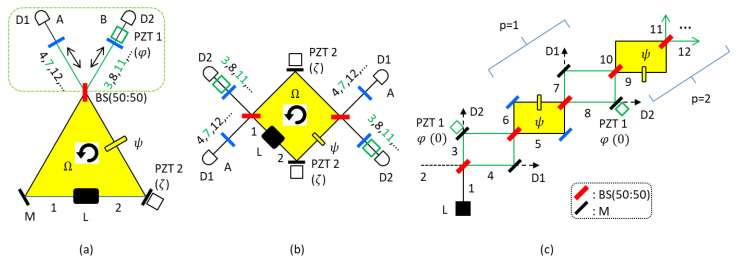
Schematic of quantum ring laser gyroscope (RLG). (**a**) Schematic of RLG based on ring cavity of CBWs. (**b**) Unfolded scheme of (**a**). (**c**) Equivalent scheme of CBWs for one mode. L: gain medium, D: photo-detector, BS: 50/50 nonpolarizing beam splitter, M: mirror, PZT: piezo-electric transducer, Ω: rotation rate. The green dotted area with a BS is the modification for CBWs. Both φ and ζ are control parameters for the quantum ring gyroscope. The ψ is the Ω-induced Sagnac effect. Due to the inserted BS, each counter-propagating (solid and dotted arrows) field pair has both transmitted (black arrow) and reflected (blue arrows) components. The numbers are the sequence of the light propagation across the BS.

**Figure 2 sensors-22-08687-f002:**
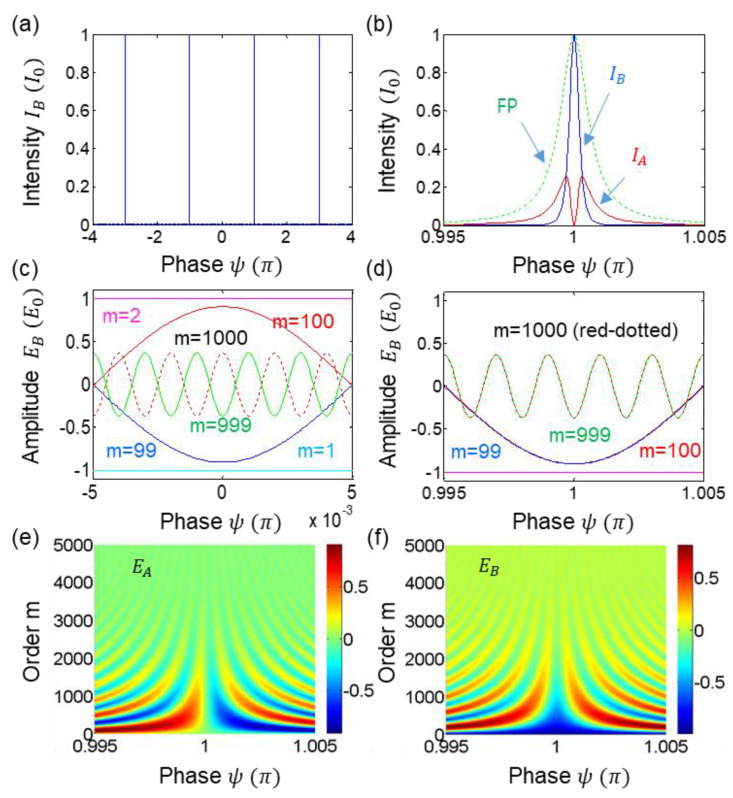
Numerical calculations for Equation (2). (**a**,**b**) Output intensities. (**c**–**f**) Output amplitudes. Amplitudes and intensities are normalized. Reflection coefficient is set at r = 0.999. The ring laser gain is not included. FP: Fabry–Perot interferometer.

**Figure 3 sensors-22-08687-f003:**
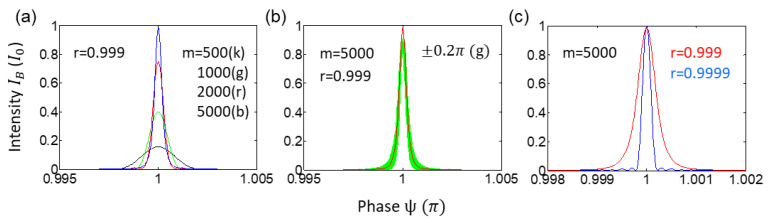
Numerical calculations for noise effects. (**a**) Cavity effects of Finesse. (**b**) Random phase noise-effect. (**c**) Gain effect (classical approach).

## Data Availability

Not applicable.

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
