# Peer review of "A Quantum Ring Laser Gyroscope Based on Coherence de Broglie Waves"

_sensors, 2022, doi:10.3390/s22228687_

Round 1

Reviewer 1 Report (New Reviewer)

The author has a list of papers, which outline the theoretical background of the sensor system. Some points needs clarification as fellows:

Line 84: Why the system has a 'high signal-to-noise ratio'?

Line 207: 'The given phase noise is assumed random within ±0.'. Why ±0.2π is assumed?

Line 253: 'three times greater than that of the RLG limit'. As there are assumptions in noise analysis. We can not get a solid conculsion about resolution enhancement.

How large the quantum noise will be in this system? What will be the quantum resolution quantitively?

Author Response

Reviewer 2 Report (New Reviewer)

Comments on the content of the article.

The author of the article claims that the article proposes a new type of the ring laser gyroscope - Quantum Ring Laser Gyroscope. The proposed type of gyroscope is shown in Fig.1. The diagram in Fig.1 is an upgraded Sagnac interferometer in which the light source is transferred inside a closed optical circuit. At the same time, due to the installation of additional mirrors, interference in this scheme becomes multibeam, i.e. the interferometer turns into a cavity (resonator). However, this resonator does not become a ring laser, since the laser assumes the presence of an active medium (amplifier) located inside the resonator. Instead of the active medium in the proposed scheme, a laser light source is located inside the resonator.

The ring laser provides the generation of two counter propagating waves, there is no generation in the proposed scheme, but instead there is a counter propagation along a closed contour of two waves from a radiation source with multiple round trip of the contour. At the same time, there is no concept of the natural frequency (eigenfrequency ) of the resonator. The waves propagating the contour in the proposed version can have any frequency.

Thus, everything is mixed up in the paper. This is not a ring laser, but a ring resonator, but there are no phase conditions of the resonator (an integer number of wavelengths is located at the length of the resonator or the phase shift after trip around resonator is equal to the integer 2π radians), so the proposed scheme cannot be considered either as a ring laser or as a ring resonator. The scheme should be considered as a modification of the Sagnac interferometer, already presented by the author in [18]. The results obtained in this work practically do not differ from the results of [18].

2. If you need to compare two optical gyroscopes by sensitivity, then this is usually done by evaluating the minimally distinguishable changes in the rotation rate.

Round 2

Reviewer 2 Report (New Reviewer)

 Response 1:

 As reviewed in refs. [30,31], an embedded laser in an optical cavity forms the ring laser gyroscope (RLG), …

Comment

Optical cavity of ring laser never has laser, it has active medium (gain medium), which cannot be considered as source of light (laser). It is only amplifier. Look at the Fig.6 in [30]. There is cathode and two anodes in triangle ring cavity. These cathode and anodes provide electrical discharge in He-Ne medium and due to it we have inversion of population in Ne atoms and this medium becomes amplifier. The cavity (resonator) plays the role of positive feedback due to which the amplifier becomes a generator and the generation of two counter waves occurs. A resonator with an amplifier becomes a laser (ring laser).

It is principal mistake of your consideration that you think that the optical cavity has embedded laser.

Author Response

This manuscript is a resubmission of an earlier submission. The following is a list of the peer review reports and author responses from that submission.

Round 1

Reviewer 1 Report

The author proposes that coherence de Broglie waves (CBWs) can overcome practical limitations of photonic de Broglie waves (PBWs) such as the difficulty in generating higher order entangled photons and that they cannot be generated deterministically. Based on the asymmetric interferometer implementation the author proposed in previous work, a quantum ring laser gyroscope is proposed.  

The analysis in the paper is linear, but ring laser gyros (RLGs) are inherently nonlinear systems because of the gain medium. The author published a previous paper on a Sagnac interferometer implementation of this idea, for which the linear analysis should be sufficient. But it is not straightforward to extend this to an RLG where issues such as gain competition and lock in may occur. For example, the proposed quantum RLG in Fig. 1 mixes the two directions. In an RLG any mixing between the directions results in deadband where the system becomes insensitive to changes in rotation rate. Additionally it’s not clear if there is any benefit without an analysis of the noise. The author claims the proposed RLG would be “environmental noise immune” like the conventional RLG. Although conventional RLGs are common mode, they are not immune to noise. Further, in Fig. 1, the two paths outside the yellow region will contribute phase shifts but are not common path.  

Author Response

Response to Reviewer 1:

Comments:

The author proposes that coherence de Broglie waves (CBWs) can overcome practical limitations of photonic de Broglie waves (PBWs) such as the difficulty in generating higher order entangled photons and that they cannot be generated deterministically. Based on the asymmetric interferometer implementation the author proposed in previous work, a quantum ring laser gyroscope is proposed.  

  1. The analysis in the paper is linear, but ring laser gyros (RLGs) are inherently nonlinear systems because of the gain medium. The author published a previous paper on a Sagnac interferometer implementation of this idea, for which the linear analysis should be sufficient. But it is not straightforward to extend this to an RLG where issues such as gain competition and lock in may occur. For example, the proposed quantum RLG in Fig. 1 mixes the two directions. In an RLG any mixing between the directions results in deadband where the system becomes insensitive to changes in rotation rate.
  2. Additionally it’s not clear if there is any benefit without an analysis of the noise. The author claims the proposed RLG would be “environmental noise immune” like the conventional RLG. Although conventional RLGs are common mode, they are not immune to noise. Further, in Fig. 1, the two paths outside the yellow region will contribute phase shifts but are not common path.  

Response to the Comment by Reviewer 1:

  1. Regarding the fundamental difference, the previous work (ref. 18) is for passive ring gyroscope, whereas the present paper is for an active one. For ref. 18, there is a strong coherence relationship between two paths in the MZI. I will call it one-mode coherence originated in the same laser input. On the contrary, the active ring gyroscope has two independent modes of coherence in both forward and backward directions, as denoted by ‘1’ and ‘2’ in Fig .1(a). Conventional active ring gyroscopes have already dealt with this matter, independently (see Fig. 4 of ref. 30). By the way, the same physics of independence between two-mode coherences has also been studied in the name of Sorkin parameter or Born’s rule in quantum mechanics in a multi-input and multi-output interferometer. In an optical cavity, the gain effect directly relates with the sensitivity with an enhanced SNR. However, this gain does not directly affect the phase resolution. Both sensitivity and resolution are of course independent parameters in sensing technologies. Although the phase resolution in the ring cavity is primarily limited by Finesse of the cavity, the optical loss inside the cavity should affect the contrast, resulting in reduced resolution due to less collections of high order reflected lights.

Unlike the passive model of ref. 18, the present scheme relates with two-mode CBWs. Thanks to the optical cavity, the phase difference between these two modes must be fixed due to the shared path length of the cavity. Figure 1(c) has been revised to clarify about the one mode of the laser L in Fig. 1(a). In an active RLG, the dithering problem has already been solved (see ref. 30). This dithering solution should be applied to the present CBW-based quantum ring gyros in the same way because of the same cavity physics. The dithering is not due to two mode lights but due to the back-scattering light coupled to its original one by the imperfect mirror (see Section 3.2.1 of ref. 30). In the present paper, the main issue is not either sensitivity or dithering but enhanced phase resolution beyond the classical limit of RLG. The ring cavity design with an additional part (green dotted) in Fig. 1(a) is the novel part of the present quantum version, where this additional part is the source of quantum superposition among infinite number of ordered CBWs. The active version of the quantum ring gyro is an extension of the passive version of ref. 18, where the enhanced SNR by gain effects may practically enhance the phase resolution via the contrast due to increased intensities in the higher order CBWs. In principle, nonlinearity of the gain effects does not contribute to the phase resolution at all. For general readers, Fig. 1(c) is modified and Fig. 3 is added with suitable texts.

  1. The noise immune is deleted to avoid any unnecessary misunderstanding. However, the environmental noise immunity in Sagnac interferometer has been experimentally demonstrated for quantum sensing (see Phys. Rev. A 73, 012316 (2006); ref. 32). Moreover, double unitary transformation for the quantum ring gyro in Fig. 1(a) greatly suppresses usual MZI phase noises due to negligibly small relative MZI phase difference between the forward and backward propagations (see Sci. Rep. 11, 1900 (2021); ref. 33). To support this, additional references of ref. 32 and 33 are added. Due to this double unitary transformation, the noise issue is not the same as the common ring gyros, either. This is the meaning of environmental noise immunity of the present scheme. To help general readers, Fig. 3 is added with two paragraphs in p. 6.

Reviewer 2 Report

The paper proposes a new scheme for an optical gyroscope. A typical gyro would have active media inside an optical resonator, such as a commercial gyro from Honeywell.

The paper considers a different layout: the output coupler is replaced by a Michelson interferometer. I struggle to see any particular advantages of the scheme compared to the standard one though because do not see any quantum entanglement in the system.

The author refers to the coupled Mach-Zender interferometers as a source of improvement. I think there might be some quantum improvements if quantum states are injected into the system. However, Fig. 1 shows a typical laser medium unless I read the figure wrong. The analysis does not include the laser medium either.

Therefore, if the laser medium creates coherent states of light that propagate through a set of passive optics, I do not see how the quantum noise could improve: where does the entanglement come from?

It would be nice to show the expected quantum sensitivity in units of rad / sqHz for similar optical powers in standard gyros and the one proposed in the paper.

Author Response

Response to Reviewer 2:

Comments:

The paper proposes a new scheme for an optical gyroscope. A typical gyro would have active media inside an optical resonator, such as a commercial gyro from Honeywell.

  1. The paper considers a different layout: the output coupler is replaced by a Michelson interferometer. I struggle to see any particular advantages of the scheme compared to the standard one though because do not see any quantum entanglement in the system.
  2. The author refers to the coupled Mach-Zehnder interferometers as a source of improvement. I think there might be some quantum improvements if quantum states are injected into the system. However, Fig. 1 shows a typical laser medium unless I read the figure wrong. The analysis does not include the laser medium either.
  3. Therefore, if the laser medium creates coherent states of light that propagate through a set of passive optics, I do not see how the quantum noise could improve: where does the entanglement come from?
  4. It would be nice to show the expected quantum sensitivity in units of rad / sqHz for similar optical powers in standard gyros and the one proposed in the paper.

Response to the Comments by Reviewer 2:

  1. First of all, the present scheme of quantum ring gyro is based on CBWs (coherent de Broglie waves) in a double unitary transformation of MZIs via an intermediate coupler (dotted region of Fig. 1(a)). The role of the intermediate MZI is critical to induce ordered quantum superposition between consecutive MZIs (yellow regions of Fig. 1(c)). The physics of passive ring gyro based on CBWs has already been discussed in the previous work (see ref. 18). The physics of conventional ring laser gyro is basically a two-mode ring resonator whose physics is the same as Fabry-Perot (see ref. 30). The gain medium in an active ring gyro does not affect the phase resolution in principle, as shown in ref. 30. The direct benefit of the gain medium is the enhanced SNR, contributing to high sensitivity. The direct benefit of the ring cavity is the enhanced phase resolution.

The physics of the present quantum RLG is completely different from conventional RLG, where the additional part of the dotted region in Fig. 1(a) plays a critical role to induce quantum superposition between adjacent MZIs (yellow regions). This quantum superposition by the intermediate coupler creates a nonclassical feature of phase resolution, which is beyond the diffraction limit in Fabry-Perot, as numerically demonstrated in Fig. 2(b). The related analysis has already been well explained in Results. This nonclassical feature of enhanced phase resolution by CBWs has nothing to do with conventional entanglement or squeezed state, where the common interpretation of entanglement is based on the particle nature of quantum mechanics. The CBWs are based on the wave nature of quantum mechanics as explained in Introduction (see refs. 16 and 17).

  1. Unlike conventional entangled or squeezed state-based quantum sensing, present paper is based on CBWs: refs. 16-18. As is well known, the enhanced phase resolution in a Michelson interferometer has already been observed in LIGO gravitational wave detections using squeezed light (Nature Photon 7, 613 (2013)). The squeezed state-based sensitivity enhancement is a factor of two at most. Unlike squeezed light, the order of entangled state can give more enhanced sensitivity as well as phase resolution by increasing entangled photon numbers in a form of N00N state, according to the Heisenberg limit. However, higher-order N00N state generation is extremely difficult and thus impractical for quantum sensing. On the contrary, the present CBW-based quantum RLG does not need such intensity-entangled particles or squeezed light. Instead, the ordered quantum superposition between consecutive MZIs (see Fig. 1(c)) gives an essential role for the nonclasical feature beyond the diffraction limit, as numerically demonstrated in Fig. 2(b) as well as ref. 18. The present scheme of Fig. 1 has nothing to do with conventional entangled or squeezed states for quantum sensing but CBWs for enhanced phase resolution. Figure 1(a) is not a conventional MZI, either, where the added intermediate part of the green-dotted region is essential to generate CBWs via quantum superposition between MZIs, as shown in Fig. 1(c) and Fig. 2(b).
  2. Using common laser fields, CBW is created and applied for the present quantum RLG, as shown in Fig. 1. There is no N00N state or squeezed light in Fig. 1. This is the novelty of the present paper, which is completely different from both entanglement-based quantum sensing and Fabry-Perot based conventional RLG. The enhanced resolution of the present model is shown in Fig. 2(b), which shows a nonclassical feature of phase resolution beyond the conventional RLGs. The enhanced phase resolution originates in CBW superposition, where CBWs have linearly doubled wavelengths (see refs. 16-18). The quantum noise is suppressed by superposition of CBWs in the cavity as shown in Fig. 2(b). To help general readers, Fig. 3 is added with related texts in the revised version.

There are two independent parameters in sensing. One is sensitivity and the other is resolution. The present paper is not for the sensitivity but for the resolution. These two terms are independent parameters. The present paper is for the enhanced phase resolution compared to the conventional RLG. Thus, mentioning sensitivity is beyond the scope of the present paper. In a RLG, enhanced sensitivity is obvious due to the optical gain via higher SNR. Thus, at the present stage, it is not appropriate to add quantum sensitivity, which is not the main issue of the proposed quantum RLG, either.

Round 2

Reviewer 1 Report

One concern is that the large coupling between the two directions introduced by the 50/50 beamsplitter could result in any Sagnac phase shift (within the yellow portion) being averaged out. This might not be the case in the interferometer analysis owing to the asymmetry and portions outside the yellow, but these portions are not common mode in the ring laser and would suffer from random phase fluctuations. In addition, the large coupling would likely impact the lasing threshold (resolution and sensitivity are not independent in this case), the lasing frequencies, and the phase noise, and could lead to nonlinear dynamics. While perhaps a starting point, I don't feel that the linear analysis provides a sufficient picture of the behavior of the proposed system.

Reviewer 2 Report

Dear authors, I am sorry for this short review but I just cannot understand how the quantum noise is suppressed in your scheme. During the last review, I requested a figure which shows the spectrum of the quantum noise of a typical RLG compared to your design but the figure is still missing. Therefore, I do not understand how much you improve the quantum sensitivity. Also, I do not understand why any sensitivity improvement is expected at all. 
